# Color-Specific Recovery to Extreme High-Light Stress in Plants

**DOI:** 10.3390/life11080812

**Published:** 2021-08-10

**Authors:** Débora Parrine, Todd M. Greco, Bilal Muhammad, Bo-Sen Wu, Xin Zhao, Mark Lefsrud

**Affiliations:** 1Department of Bioresource Engineering, Macdonald Campus, McGill University, 21,111 Lakeshore Boulevard, Sainte-Anne-de-Bellevue, Québec, QC H9X 3V9, Canada; debora.parrine@imbim.uu.se (D.P.); bo-sen.wu@mail.mcgill.ca (B.-S.W.); 2Department of Molecular Biology, Princeton University, Princeton, NJ 08544, USA; tgreco@princeton.edu; 3Department of Animal Science, Macdonald Campus, McGill University, 21,111 Lakeshore Boulevard, Sainte-Anne-de-Bellevue, Québec, QC H9X 3V9, Canada; muhammad.bilal@mail.mcgill.ca (B.M.); xin.zhao@mcgill.ca (X.Z.)

**Keywords:** abiotic stress, photosystem II, extreme high light, NPQ, LED

## Abstract

Plants pigments, such as chlorophyll and carotenoids, absorb light within specific wavelength ranges, impacting their response to environmental light changes. Although the color-specific response of plants to natural levels of light is well described, extreme high-light stress is still being discussed as a general response, without considering the impact of wavelengths in particular response processes. In this study, we explored how the plant proteome coordinated the response and recovery to extreme light conditions (21,000 µmol m^−2^ s^−1^) under different wavelengths. Changes at the protein and mRNA levels were measured, together with the photosynthetic parameters of plants under extreme high-light conditions. The changes in abundance of four proteins involved in photoinhibition, and in the biosynthesis/assembly of PSII (PsbS, PsbH, PsbR, and Psb28) in both light treatments were measured. The blue-light treatment presented a three-fold higher non-photochemical quenching and did not change the level of the oxygen-evolving complex (OEC) or the photosystem II (PSII) complex components when compared to the control, but significantly increased *psbS* transcripts. The red-light treatment caused a higher abundance of PSII and OEC proteins but kept the level of *psbS* transcripts the same as the control. Interestingly, the blue light stimulated a more efficient energy dissipation mechanism when compared to the red light. In addition, extreme high-light stress mechanisms activated by blue light involve the role of OEC through increasing PsbS transcript levels. In the proteomics spatial analysis, we report disparate activation of multiple stress pathways under three differently damaged zones as the enriched function of light stress only found in the medium-damaged zone of the red LED treatment. The results indicate that the impact of extreme high-light stress on the proteomic level is wavelength-dependent.

## 1. Introduction

Photosynthesis is the metabolic process most impacted by abiotic stress. Plants under full light absorb up to 10% of the available light, directing energy to photosynthetic electron transport [1]. The excess energy must be dealt with through photoprotection mechanisms in order to protect the photosystems from photoinhibition. It is known that photoinactivation is a wavelength-dependent mechanism, and has been studied to some extent in various action spectra, and an extensive review of this topic has been reported by Zavafer et al. [2]. However, there is a lack of knowledge on how extreme light intensities can impact the proteome of plants under different wavelengths beyond the levels of photoinhibition. There is also little information on how physiological parameters such as nonphotochemical quenching and photosynthetic rate are impacted by extreme high-light treatments.

We recently reported on a high-light-induced stress treatment utilizing a narrow red spectrum LED light under a 5000 W m^−2^ intensity [3]. The effect of the treatment was studied in depth by using an isobaric-labeled proteomics strategy (iTRAQ). We identified key proteins involved in the long-term response to the stress condition: PsbS, PsbR, PsbH, and Psb28, and reported the differential expression of these components of the PSII and the oxygen-evolving complex (OEC). These proteins present different roles in the photosynthesis reactions, from non-photochemical quenching (NPQ) enhancement to complex assembly facilitators. Studies have shown levels of NPQ in wildtype Arabidopsis [4] and tomato plants [5,6] under different levels of light-stress. NPQ measurements can present low values (<0.1) [7], particularly during the induction phase (first minutes after the saturating light is emitted), although here we report on a three-fold NPQ increase, during the induction phase, in tomato plants treated with blue when compared to red extreme high-light treatment.

One mechanism for NPQ regulation is performed by the photosystem II subunit S (PsbS) protein. PsbS under protonation, along with the zeaxanthin formation on the xanthophyll cycle, leads to PSII antenna conformational modifications, resulting in quenching of the PSII antenna excitation energy [8]. The role of PsbS as a sensor of overexcitation has already been suggested [9], along with its role in photoprotection [4,10]. It has been shown that the insertion of the violaxanthin de-epoxidase npq4 (PsbS gene) and zeaxanthin epoxidase genes from *Arabidopsis* into tobacco caused an increase of 15% in final dry mass [11]. The productivity increase was found to be caused by the acceleration of NPQ relaxation duration in fluctuating light. The 10 kDa protein PsbR has been suggested to be the PsbP and PsbQ docking protein for the oxygen-evolving complex formation [12]. PsbR and PsbQ were found to be involved in the organization of PSII elements, and, when absent in *Arabidopsis*, the short-term adaptive mechanism was impaired [13]. The Psb28 protein has been linked to high-light stress conditions at high temperature when PSII is damaged, and increased PSII turnover is necessary [14]. Recently, a cross-linking mass spectrometry-based study detected direct Psb28 binding to cytochrome *b*_559_ [15], suggesting a relationship between the poorly characterized Psb28 protein and the cytochrome *b*_559_. The PsbH protein is part of the PSII complex core on the acceptor side of the electron transportation, playing a role in PSII stabilization [16]. The cyanobacterium *Synechocystis* 6803 psbh^−^ mutant presented PSII electron transfer impairment between quinones Q_A_ and Q_B_, and higher sensitivity to photoinhibition under high light. In *Arabidopsis*, PsbH has been shown to be important for CP47 accumulation, a component of the inner antenna complex that directs the energy at the outer antennae to the reaction center [17].

In this work, we investigate the impact of an extreme high-light treatment on tomato leaves caused by a blue LED at the proteomics level. Differences between the recovery from extreme high-light intensity treatment from blue and red LEDs are expected due to the disparate pathways triggered by them. How the extreme level of irradiance can affect these known mechanisms is unknown. In this study, we demonstrate through photosynthetic performance, protein, and mRNA measurements the damaging impact on photosynthesis under the two wavelengths with different time spans. To complement our analysis, a mass spectrometry-based proteomics approach was utilized to generate a broad characterization of the protein abundance patterns in the leaves. Lastly, the functional role of the differentially expressed proteins was assessed to compare the plant long-term stress response under different LED treatments.

## 2. Materials and Methods

### 2.1. Plant Variety

Tomato (*Solanum lycopersicum*) variety Heinz 1706 was provided by HeinzSeed (Stockton, CA, USA). Heinz 1706 has been genetically sequenced [18], the genome has a haploid chromosome number of 12 containing 900 Mb and 35,000 protein-coding genes (genes or transcripts containing an open reading frame), and the genome annotation is still in progress.

### 2.2. Plant Growth and Sampling

The tomato seeds were planted and grown hydroponically in rockwool (Grodan A/S, Hedehusene, Denmark). Ten plants were incubated under cool-white fluorescent bulbs (4200 K, F72T8CW, Osram, Wilmington, NC, USA) in a growth chamber (TC30, Conviron, Winnipeg, MB, Canada). The environmental conditions in the chamber were controlled at 50% relative humidity (RH), 25 °C light/dark temperature, ambient CO_2_ concentration, and a 16 h photoperiod with an irradiance level of 55 W m^−2^ (approximately 250 µmol m^−2^ s^−1^). Fresh Hoagland nutrient solution was provided every other day. The Hoagland solution contains 6.5 mM KNO_3_, 4 mM Ca(NO_3_)_2_, 2 mM NH_4_H_2_PO_4_, 2 mM MgSO_4_, 4.6 µM H_3_BO_3_, 0.5 µM MnCl_2_, 0.2 µM ZnSO_4_, 0.1 µM (NH_4_)_6_Mo_7_O_24_, 0.2 µM CuSO_4_, and 45 µM FeCl_3_ [19]. After a growing period of 30 days, leaves of tomato plants were placed under a blue LED light (470 nm, LXML-PR01-0500, Philips-Lumileds, San Jose, CA, USA) with an average irradiance level of approximately 21,000 µmol m^−2^ s^−1^ on a ~1 cm^2^ spot in the center of a mature leaf for 5 min. After the LED treatment, plants were returned to the growth chamber for a 10-day period. Each group of 10 treated leaves was then collected as one biological sample, to eliminate individual variances. The leaves were dissected, and the areas corresponding to the light-treated zone (Burned), adjacent (Limit), and rest of the leaf (Regular) were separated (Figure 1A), and the remaining parts were discarded. Plant tissues were kept under −80 °C before protein extraction. The control plant group was kept in the growth chamber during the full experiment without intense irradiation, and the experiment was replicated three times.

### 2.3. Light Treatment

Tomato leaves were set under the blue LED lights at approximately 2.5 cm distance, where light intensity was stable at 5000 W m^−2^. The photon flux densities of the 470 and 655 nm LED light were approximately 21,000 and 25,000 µmol m^−2^ s^−1^, respectively. Light intensity was measured by a spectroradiometer (PS-300; Apogee, Logan, UT, USA). A filtering lens with known transmitted percentages was placed on the spectroradiometer and to attenuate the high light, as described in Wu and Lefsrud [20]. Briefly, the LED assembly was mounted on a water jacket (ST-011, Guangzhou Rantion Trading Co., Zhejiang, China) and attached to a cluster concentrator optics (25 mm focal length, No. 263, Polymer Optics, Wokingham, Berkshire, UK). All the rays from diodes were collimated by the cluster concentrator optic, resulting in a small focal spot of 12 mm in diameter of all the rays from diodes on the LED assembly being generated by the cluster concentrator optic. An isotemp bath circulator (4100R20, Fisher Scientific, Hampton, NH, USA) was used to circulate a 0 °C coolant in the water jacket. Leaf temperature was measured in three biological replicates as reported by Dixon and Grace [21] with copper constantan thermocouples (type T, 0.03 mm, Omega Engineering Canada, St-Eustache, QC, Canada). The temperature was recorded every second for 15 min total, including 5 min before and after the light treatment as well as during the 5 min wavelength treatment. The thermocouples were placed on the surface of the leaf using glue extracted in chloroform from sellotape.

### 2.4. Fluorescence and Analysis of Measurements

Chlorophyll (Chl) fluorescence measurements were performed using a leaf chamber fluorometer (LI-6400-40, LI-COR Inc., Lincoln, NE, USA). Measurements were done in triplicates to obtain the photochemical efficiency of PSII (Fv/Fm) of dark-adapted leaves. The measurements were performed following previously established guidelines [22]. Briefly, a modulated red radiation of approximately 2 μmol m^−2^ s^−1^ was used to excite fluorescence by using a frequency and a pulse width of 20 kHz and 3 µs, respectively. An approximately 8000 μmol m^−2^ s^−1^ saturating radiation pulse of 0.8 s was utilized. The open PSII center (F0) minimum Chl fluorescence and the closed PSII center maximal Chl fluorescence values were obtained after a 20 min dark-adaptation period. After, leaves were irradiated continuously, and the steady-state fluorescence (Fs) was determined. A new 8000 μmol m^−2^ s^−1^ saturating pulse was emitted to obtain the maximal fluorescence of the light-adapted state (Fm′). Then, the actinic PPFD was turned off and a far-red (740 nm) light was used to measure the minimum fluorescence of the light-adapted state (F0′). The obtained values were used to calculate the following: (i) Fv/Fm = (Fm − F0)/Fm, the maximum dark-adapted PSII photochemical efficiency, and (ii) ΦPSII = (Fm′ − Fs)/Fm′, the effective light-adapted photochemical efficiency.

### 2.5. Net Photosynthesis Rate (Pn)

The net photosynthesis rate (Pn) was determined using the leaf chamber fluorometer of the portable photosynthesis system (LI-6400, LI-COR, Licoln, NE, USA) on a fully expanded tomato leaf. The light condition was set as 100 µmol m^−2^ s^−1^, with an equal amount of 470 and 630 nm light. The environmental factors controlled during the measurement were block temperature (23 ± 1 °C), CO_2_ concentration (400 ± 1 μmol mol^−1^), and relative humidity (RH, 50–60%). The measurements were taken every 4 s for 15 min and replicated in three different plants for each wavelength treatment and the control. 

### 2.6. Protein Extraction and Digestion

A 100 mg sample was processed for protein extraction and digestion as previously described, with modifications [23]. Briefly, leaves were ground by mortar and pestle in liquid nitrogen. The powder was solubilized in a buffer containing detergent (100 mM Tris-HCl/4% SDS pH 8). The sample was boiled for 5 min before it was sonically disrupted in an ice bath (40% amplitude, 10 s/10 s on/off cycles) for a total duration of 2 min. The crude extract was processed by centrifugation at 4 °C for 10 min at 4500× *g*. The sample was adjusted to 10 mM TCEP (Tris(2-carboxyethyl)phosphine) (Sigma-Aldrich Canada, Oakville, ON, Canada), instead of DTT used in the cited protocol. TCEP was used, as it is a more powerful reducing agent and more resistant to oxidation. Non-protein contaminants were removed by 20% trichloroacetic acid (TCA) precipitation and washed in ice-cold acetone [24]. The proteins in the pellet were denatured in 8 M urea in Tris-HCl pH 8.0 for 30 min at room temperature, and sonication pulses of 10 s and 20 s on/off during 5 min in cold water were applied to cells to solubilize the proteins and avoid SDS precipitation. A fraction of the sample was diluted to 1 M urea for protein concentration measurement by BCA assay (Pierce Biotechnology, Waltham, MA, USA). The reduction of the proteins was carried out with the adjustment of the samples to 20 mM TCEP-HCl. Cysteines were blocked, and disulfide bridges were prevented with 20 mM iodoacetamide (IAA) at room temperature for 30 min in the dark. Proteins were digested with modified sequencing-grade trypsin (Promega, Madison, WI, USA), in a 1:50 ratio of trypsin to initial biomass for 12 h incubation at 37 °C. An acidic solution (200 mM NaCl, 0.1% formic acid) was added to stop the trypsin reaction. Trypsin and undigested proteins were removed by a 30 kDa MWCO Amicon ultra-centrifugal filter (Millipore Sigma, Burlington, NJ, USA). Peptides were desalted using a centrifugal column (Sep-Pak Plus C-18, Waters Limited, Mississauga, ON, Canada) before peptide quantification (Pierce Quantitative Colorimetric Peptide Assay, Thermo Fisher Scientific, San Jose, CA, USA) and storage at −80 °C.

### 2.7. Liquid Chromatography–Mass Spectrometry (LC−MS/MS)

A multi-dimensional protein identification technology (MudPIT) approach was performed to obtain a label-free shotgun proteomics analysis [25]. A high-performance separation of the peptides was obtained by using a 2D-LC separation coupled online with the mass spectrometer (LTQ XL, Thermo Fisher Scientific, San Jose, CA, USA) as previously described [26]. Approximately 60 µg of peptides from three biological replicates of each condition were bomb-loaded through a cell-pressure chamber into a biphasic column packed with ~5 cm of strong cation exchange (SCX) resin and ~5 cm of C18 reversed-phase (RP) material (Luna 5 μm 100 Å and Aqua 5 μm 100 Å, respectively; Phenomenex, Torrance, CA, USA). The packed column was washed through the cell-pressure chamber for 60 min with 0.1% formic acid in H_2_O (MS-grade Optima, Thermo Fischer Scientific, Waltham, MA, USA) to remove salts and impurities. Peptide spray was generated by a front column containing an integrated nanospray emitter tip (100 μm i.d., 360 μm o.d., 15 μm i.d. tip, New Objective, Woburn, MA, USA) loaded with ~15 cm of C18 material and in line with the back column. Liquid chromatography was carried out by an HPLC Surveyor Plus (Thermo Scientific, San Jose, CA, USA) at a ~300 nL min^−1^ flow rate at the nanospray tip. 

A 12-step gradient (24 h analysis duration) containing salt pulses was utilized to elute the peptides from the column in a nanoESI-MS/MS approach as previously described [27,28]. The gradients contained an increasing ammonium acetate concentration (0–500 mM), followed by a reverse-phase gradient elution of up to 2 h duration. The data-dependent acquisition parameters were specified in Xcalibur (v.2.0.7 SP1 Thermo Fisher Scientific, Waltham, MA, USA), where MS/MS fragmentation by collision-induced dissociation (CID, 35% energy) was performed on the 5 most intense precursor ions detected in the preceding full MS scan. Two microscans were averaged for every full MS and MS/MS spectrum for both full and MS/MS scans a 3 *m*/*z* isolation width was allowed. Dynamic exclusion was enabled with a repeat of 1 for 60 s.

### 2.8. Database Searching and Statistical Analysis

Thermo RAW files were used to extract the MS/MS spectra, which were searched against a database containing the target and reverse peptide sequences of *Solanum lycopersicum* (UNIPROT, proteome UP000004994) containing 33,952 entries, and common contaminants (cRAP v. 2012.01.01, obtained from www.thegpm.org/crap (accessed on 5 August 2021)). The MSAmanda 2.0 [29] algorithm was used for protein identification through the software Proteome Discoverer v.2.1.1. (Thermo Fischer Scientific, Inc., Waltham, MA, USA). A search against the reverse sequence (decoy) tomato database was performed to allow estimation of the global false discovery rate (FDR). The search parameters were set to a maximum of 2 missed cleavages, with a parent ion and fragment tolerance of 2.0 Da and 0.4 Da, respectively. Methionine oxidation (+15.99 Da) was set as variable modification and carbamidomethylation of cysteines (+57.05 Da) as a static modification. Proteins were filtered to contain ≥2 unique peptides and ≥4 PSM in the 2 biological replicates of each sample (4 conditions × 2 biological replicates), and only proteins found in 4 out of 8 samples were further analyzed. 

### 2.9. Bioinformatics

The results were imported into Perseus [30], where the NSAF [31] values were normalized across samples to account for differences in global protein abundance. Proteins with a fold change of 2.5 were considered significantly differentially abundant. A semiquantitative comparison of the protein abundance across all samples was performed as follows. A normal distribution of data was obtained by log_2_-transforming the expression values (NSAF) after their normalization. Functional annotations of the identified proteins were obtained via ClueGO [32]. The network reflects the level of relationship amongst the GO terms assigned to the proteins used as input. The nodes reflect the statistical significance of their assigned terms by their size. Kappa statistics were used to calculate the degree of connectivity (edges), and the definition of functional groups, similar to that described in Huang et al. [33].

### 2.10. Quantitative Reverse Transcription PCR (RT-qPCR)

The total RNA from leaf samples was extracted with the RNeasy Plant Mini kit (Qiagen, Hilden, Germany). The QuantiTect Reverse Transcription kit (Qiagen) was used to synthesize the cDNA as presented in the manufacturer’s protocol. RT-qPCR primers were designed using the online tool Primer-BLAST from the National Center for Biotechnology Information (NCBI) (Table 1). The mixed solution of RT-qPCR reaction contained Platinum SYBR Green qPCR SuperMix-UDG with ROX (Invitrogen, Waltham, MA, USA), reverse and forward primers mix (4.28 μM), and 20-fold-diluted cDNA template. All reactions were performed on a CFX Connect Real-Time PCR system (Biorad, Hercules, CA, USA). The reaction conditions were 10 min at 95 °C, followed by 40 cycles of heating at 95 °C and annealing at 60 °C for 15 and 60 s, respectively. Melting curves were carried out in each RT-qPCR to verify single-product amplification. The relative level of gene expression was calculated with the Livak method (2^−ΔΔCt^). The genes protein phosphatase 2A catalytic subunit (*PP2Acs*) [34] and clathrin adaptor complex subunit (*clat*) [35] were used as the reference genes. Measurements were recorded from three technical and three biological replicates for each experimental condition. 

### 2.11. Statistical Analysis

The statistical analysis was carried out with the values of the biological triplicates of leaf photosynthesis rates, chlorophyll fluorescence, and an additional three technical replicates of each biological replicate for the qPCR analysis. The mean values were tested using ANOVA, and pair-wise comparisons were adjusted with Tukey’s HSD test (*p* ≤ 0.05).

## 3. Results

### 3.1. Plant Physiological Stress Measurements 

The blue- and red-light treatments (hereafter abbreviated as BLT and RLT, respectively) were compared to the level of physiological damage. The blue wavelength was chosen due to its well-characterized physiological response in plants, and the involvement of blue photoreceptors in triggering plant acclimation [36,37,38]. The data collected from the BLT was compared to the RLT results previously reported [3]. 

Like the RLT, the BLT performed in this study generated a highly dehydrated and damaged spot of ~1 cm diameter on the tomato leaves (referred to as the Burned sample) and two other zones of lower light intensity (in decreasing intensity order: Limit and Regular) (Figure 1A). We defined Day 0 and Day 10 as the time point right after the light treatment and 10 days later, respectively. After a period of 10 days, the Burned area showed symptoms of de-etiolation, the Limit area showed a slightly darker green color, and the Regular area had no change in appearance. 

The plant temperature was monitored during the high-light treatment (Figure 1B). Leaf temperature was measured by a thermocouple during the high-light treatment using blue and red. The light treatment started at 170 s and ended at 470 s (~5 min duration). The red LED temperature data used for comparison were published by Parrine et al. [3]. The temperature measurements of the BLT showed a shift in temperature of about 80 °C after 120 s of the experiment (~2 min). The RLT generated higher temperatures in general, and resulted in a faster temperature rise. Only by the end of the experiment did the BLT reach a similar temperature to the RLT of 94 °C and 99 °C, respectively.

### 3.2. Impact of High-Light Induced Heat Stress in Photosynthesis Efficiency

Plant-stress indicators were measured to determine the extent of the physiological damage caused by the BLT and RLT, as well as whether the leaf could recover its photosynthetic efficiency. A comparison of the measurements of the net photosynthesis rate (Pn), the maximum quantum efficiency of PSII photochemistry, and the non-photochemical quenching values of the BLT and RLT are shown in Figure 2. 

The Fv/Fm parameter is an indicator of the maximum quantum yield of PSII. In healthy plants, the maximum Fv/Fm value is ~0.83 [22], agreeing with the experimental value obtained for the control plants (0.81 ± 0.003). The plants treated with BLT presented a slightly higher Fv/Fm (not statistically significant) on Day 10 compared to Day 0. The measured Fv/Fm values from RLT did not present a change from Day 0 to Day 10. In addition, the difference between the RLT and the BLT on Day 0 and Day 10 was not significant. Considering the treatments separately, when compared to the control at Day 0 and Day 10, BLT and RLT presented a statistically significant difference (>10-fold). Although similar values of Fv/Fm for RLT and BLT on Day 0 and Day 10 were obtained, as expected, a differential impact between the light treatments was observed in the NPQ parameter (Appendix A). The NPQ results showed a statistically significant difference between the RLT and the BLT on Day 0. The NPQ for BLT was three-fold higher on Day-0 when compared to the RLT values. It is important to note that the NPQ measurements reported here were taken during a small period of time in the induction phase, and therefore the values should be compared to early data points of NPQ curves found in the literature [40] (i.e., a compilation of NPQ curves recorded from tomato plants). Still, the NPQ values obtained from in the control plants were lower than numbers reported in the literature and should be taken with caution.

### 3.3. Photosystem II-Related Protein Abundance Comparisons

Proteomics analysis of the tomato leaves after a 10-day period after the high-light treatment with the blue LED was performed. This time point was chosen rather than Day 0 since the samples after the extreme high-light-induced stress would have yielded, in the majority, proteins degraded by the level of light and heat (such as NPQ). The 10-day period allows for the observation of the functions activated after the damaged central pathways were restored. We then compared our proteomics analysis of the BLT with our previous observations of protein changes under red-light treatment (RLT) [3]. In our study of the RLT, we reported on a considerable unique pool of proteins showing an increased abundance in the Burned sample when compared to the Regular and Limit samples. Amongst the proteins presenting functions in photosynthesis reactions, Psb28, PsbH, PsbR, and PsbS had, respectively, a 1.08, 1.65, 1.11, and 1.32 log_2_(-fold change) increase in abundance when compared to the control (Table 2). Due to their differential abundance patterns and their importance in photosynthesis, these proteins were chosen for further investigation. 

To determine whether the relative abundances of the proteins of interest followed the same pattern under a different wavelength, a proteomics experiment was carried out on the plant leaves treated with the BLT. In this study, a global proteomics approach was performed, and the results were compared to the RLT dataset. Although the relative abundance values from RLT and BLT datasets were calculated using different quantitative proteomics techniques (ion intensity for the RLT quantification and normalized spectral counting, NSAF, for the BLT quantification) and therefore should not be directly comparable in quantitative terms, a comparison of their representativeness amongst the other proteins of their datasets and between differential functional protein classes is still valid. Therefore, specific significance cut-offs for the RLT and BLT datasets were applied due to the different mass spectrometers used for the proteomics analysis.

In the results of the protein quantification of Psb28, PsbS, PsbH, and PsbR in the BLT treatment, in contrast with the protein measurements of RLT, the fold change in NSAF indicated a slight trend of downregulation for the proteins in the Burned sample (Table 2), though these changes were not considered significant, since the defined significance cut-off for (log_2_)NSAF-based differential abundance was >2.5. The values of abundance of the proteins of interest measured in the Limit and Regular samples of BLT were higher when compared to the Burned sample, presenting trends to high abundance in PsbS in the Limit, and PsbR in the Limit and Regular. Psb28 and PsbH presented similar abundance values as their control.

### 3.4. Correlation of Gene Expression with Protein Abundance

We then evaluated whether the changes in the protein abundances of Psb28, PsbH, PsbS, and PsbR were a result of regulation at the gene transcription or translation level. The results of the quantitative analysis of the mRNAs of interest by RT-qPCR in the various samples (Burned, Limit, and Control) in the two light treatments (BLT and RLT) are presented in Figure 3. Samples from the same experiments were utilized to obtain the protein and mRNA measurements.

The transcription level of the *psb28* gene was consistent across all samples, remaining equal to the control. However, the protein abundance measurement of Psb28 in the red Burned sample was significantly higher than the control.

The *psbH* transcript levels in the Limit sample from BLT and RLT were significantly lower, with the same trend for RLT at the protein level. The transcript levels were the same as the control for the Burned sample both in the RLT and BLT conditions, yet the protein abundance in the BLT was significantly higher. 

The protein and transcript levels of *psbR* in the RLT Burned sample were highly abundant, whereas in the other samples, the levels of transcript and protein were equal to the control. 

The *psbS* mRNA levels showed no statistical difference in all the treatments relative to the control, except for the Burned sample from the BLT, where a substantial eight-fold increase was detected. Although the PsbS protein measurements in the Burned sample of the RLT showed to be in high abundance, with no difference in the transcript level, interestingly, no difference was detected in the BLT Burned sample, which exhibited a greater abundance increase.

### 3.5. Other Proteins Identified in the BLT Dataset

The global proteomics analysis of BLT detected 43 differentially abundant proteins (Table 3 and Table 4). For the group of upregulated proteins, the Limit sample was the one that contained the most proteins, followed by the Regular and the Burned, with the latter containing only one (K4ATA4). The Burned sample presented the highest number of downregulated proteins, followed by the Limit and the Regular samples. To determine the functional interactions of the proteins and to obtain the role of uncharacterized proteins (usually by homology annotation), a GO term enrichment was performed through the platform ClueGO. 

The proteins found to be more abundant in the Regular sample of the BLT dataset were mostly involved in protein and macromolecular complex subunit organization, assembly, biogenesis, and metabolic process, as well as protein oligomerization and proteolysis, with protein hexamerization being the most significant function (Figure 4A). The lower-abundance proteins did not generate a network, but their molecular functions were related to catalytic, hydrolase, and peptidase activity.

The Limit sample had the upregulated proteins distributed amongst three clusters presenting the main functions: negative regulation of the cellular metabolic process, catabolic process, and cellular component organization or biogenesis (Figure 4B). The downregulated proteins had roles in metabolic and cellular processes, regulation of biological processes, and response to heat and cytokinin.

The Burned sample of the BLT dataset presented downregulated proteins with distinct functional classes compared to the other samples, which were related to three major clusters: photosynthesis, coenzyme biosynthetic process, and oxidative stress response (Figure 5). 

The protein found in significant upregulated abundance (K4ATA4: uncharacterized protein) has the molecular function of mRNA binding, and its highest homology (59.4% score) is with the zeaxanthin epoxidase protein from the pink trumpet tree (*Handroanthus impetiginosus*).

## 4. Discussion

### 4.1. Implications in the Photosynthetic Responses to High-Light Stress under Different Wavelengths

High-intensity light treatments in plants will not only cause light stress but will also induce heat production, although it has also been shown that short high-light treatments in some cases can result in improved photosynthetic performance [41]. The temperature in the tomato plant under the different light treatments was higher in plants under blue high light. Indeed, blue wavelengths cause higher photodamage when compared to other wavelengths from the visible spectrum [42,43]; therefore, a faster or more efficient trigger for photoinhibition would be expected as a strategy to avoid photoinhibition. One way to efficiently prompt photoinhibition is through the manganese-dependent mechanism. The manganese cluster has affinity to UV and the blue wavelength range, and it can trigger photoinhibition through its release from the OEC [44]. Plants under red treatment would possibly be under a higher level of unbalance in PSII, having to synthesize and assemble its components to keep their stoichiometric balance. 

The Fv/Fm parameter is an indicator of the maximum quantum yield of PSII. It measures the dark-adapted fluorescence emission variation in plants as a measurement of PSII damage, and it has recently been indicated to also be a tool for accurate stress forecasting for plants impacted by pollution [45]. By measuring the levels of chlorophyll fluorescence, an evaluation of the physical conditions of the electron transport chain and the PSII components can be performed [46]. Two measurements are necessary: F0, which is obtained when a low light that cannot drive photosynthesis is utilized, and Fm, the maximum fluorescence emitted when a saturating light pulse causes the reaction centers to be closed. Fv is obtained by calculating the difference between the maximum (Fm) and the minimum fluorescence (F0). In stressed plants, the Fv/Fm ratio is decreased, since fewer reaction centers are open. In healthy plants, the maximum Fv/Fm value is ~0.83 [22], agreeing with the experimental value obtained for the control plants (0.81 ± 0.003). Differences in Fv/Fm values between RLT and BLT could be expected, due to the blue wavelength impact in photosystem efficiency, photosynthetic electron transport, chlorophyll content, and a/b chlorophyll ratio (and chlorophyll-PSII binding proteins) [43]. Interestingly, it has been shown that a pre-treatment with phytohormone abscisic acid can assist in the recovery of Fv/Fm, NPQ, and other parameters [47].

The control values of Pn (~2.58 µmol CO_2_ m^−2^ s^−1^) agreed with values reported in the literature at 100 µmol m^−2^ s^−1^ of light [39]. Negative values of Pn indicate a higher level of respiration over photosynthesis. Higher Pn values indicate a higher acclimation of photosystems for CO_2_ fixation. A lower Pn would be expected on Day 0 compared to Day 10, since the measurements were taken immediately after the treatment, when components of the photosystem complexes would probably not yet have been recovered/de novo synthesized. Although RLT had a small decrease in Pn from Day 0 to Day 10 both in BLT and RLT, the Pn values indicate the driving of energy to respiration, rather than photosynthesis. 

The results of the mRNA analysis indicate that Psb28, PsbR, and PsbH presented mRNA values lower than protein levels. The PsbH gene is present in the chloroplast, and the regulation of chloroplast genes involved in the assembly and biosynthesis of photosystems is regulated by co-location for redox regulation (CoRR). This mechanism holds one of the current theories as to why plants retain a separate genetic system despite the energetic costs of maintaining it [48]. The *rbcL* and *psbA* genes, encoding for a subunit of Rubisco and the D1 protein, respectively, have been reported as regulated by CoRR [49]. In a more recent study, minimization of photodamage and higher stability of D1 protein has been shown to be improved by high accumulation of glycinebetaine in transgenic coda-tomato plants [50]. Mutants of overexpression and knockout of *psbs* presented differences in tolerance to UV-C and cell death when exposed to episodic red or blue light [51]. The authors also report on a better effect of blue light than red in generating tolerance to UV-C in wildtype Arabidopsis, resulting in a larger impact on gene expression.

The *psbr* gene is transcribed in the nucleus and is later transported to the chloroplast. The PsbR protein is localized in the proximity of the OEC docking [16] and is responsible for the stable assembly of PsbP, an OEC protein, to the core of PSII.

Psb28 is a binder to cytochrome *b*_559_ [15], with a role in the regulation of chlorophyll availability during PSI and II biosynthesis [52].

The difference between the mRNA and protein levels can be understood by the low Pearson correlation for protein abundances and mRNA, which can be up to 0.40 [53]. The remaining variation could be explained by the many levels of DNA/RNA/protein regulatory mechanisms, or by differences in the protein and mRNA half-lives [54]. Recently, the role of glycine-rich protein OsGRP3, members of a class of RNA-recognition motifs, has been reported to associate with ROS gene transcripts, reducing the half-life of the PR5 gene, linking glycine-rich proteins to the regulation of drought tolerance through mRNA stability [55].

### 4.2. Spatial Disparities in the Response of Leaves to High-Light Stress

The functional analysis results show a considerable difference between the function of the proteins from each sample (Burned, Regular, and Limit). The proteins identified in the Burned sample indicate that the tissue was under stress-induced senescence, in agreement with recent findings showing the link between high-light stress, leaf senescence, and the triggering of jasmonic acid biosynthesis [56]. During senescence, catabolism of macromolecules, such as membrane lipids, RNA, and proteins, and chlorophyll synthesis takes over photosynthesis [57]. The other samples, Limit and Regular, had proteins related to functions of cellular organization and energy production: pathways that are linked to tissue recovery and growth. The different active functions in the tissues are evidence of the different levels of stress response and plant development strategies.

A comparison between the protein functions found in this study (BLT) and the RLT dataset shows the differences between the responses to the treatments with red and blue LEDs. In RLT, the abundant proteins in the Burned sample were related to heat, oxidative, light, and endoplasmic reticulum stresses, whereas in the BLT, the upregulated protein was implicated in oxidoreductase activity, thus related to oxidative stress, lacking the other stress types. The downregulated proteins in the BLT indicate an undergoing senescence process on the damaged tissue. 

The active functions in the BLT Limit sample were related to catabolic processes and cell organization. These processes prevent or reduce the rate of chemical reactions and pathways by which cells undertake chemical transformations of substances. They increase the chemical reactions, resulting in compound cleavage for energy liberation. These functions are necessary for the recovery of damaged plant tissues and the increase in energy availability, possibly for growth. However, in the RLT results, the Limit sample presented similar active functions to the Burned, except for the particular active light-stress response. 

Lastly, in the Regular sample from the RLT dataset, abundant proteins had roles in general stress response, whereas in BLT, the more abundant proteins were engaged in protein hexamerization and functions related to the hydrolysis of proteins into amino acids or polypeptides, and the formation of macromolecules and proteins [58], actions that are part of the cellular organization and that restructure the cells due to the stress damage.

Taken together, the results from this study suggest that the two light treatments with the same extreme light intensity, but different wavelengths, triggered disparate functions in the plant. In the BLT, most of the proteins of PSII and OEC complexes had no change in abundance when compared to their control, although PsbS mRNA levels had an eight-fold increase. Dissimilarly, the RLT proteomics results showed an increase in the proteins from PSII and OEC while presenting low-level PsbS transcripts. 

## 5. Conclusions

The photosynthetic parameters had similar values between the BLT and RLT treatments and amongst the Day 0 and Day 10 samples. Additionally, no physiological difference in photosynthesis efficiency was recorded amongst treatments and time. The analysis of the abundance of the key proteins Psb28, PsbS, PsbH, and PsbR showed a higher concentration of these proteins in the RLT, indicating a differential recovery of the PSII and OEC complexes under different wavelength treatments. The mRNA quantification analysis suggests that regulations out of the transcriptional supervision could be involved in the control of *psb28*, *psbh*, and *psbr* genes. In addition, the high level of PsbS mRNA in the BLT could indicate a higher level of NPQ PsbS-dependent response under extreme high-light conditions. Curiously, the spatial proteomics analysis showed enrichment for the light-stress pathway only in the RLT Limit sample. The global functional analysis of the Burned proteins shows that higher damage was caused by the BLT when compared to the RLT, resulting in stress-induced senescence in the higher damaged zone of the leaf. Our study demonstrated that as in natural conditions, in extreme light conditions, light wavelengths impact the plant recovery in a different manner. New studies aiming at the exploration of the mechanisms involving PsbS regulation and its role in inducing NPQ are needed to better clarify the responses to extreme high light under different wavelengths.

## Figures and Tables

**Figure 1 life-11-00812-f001:**
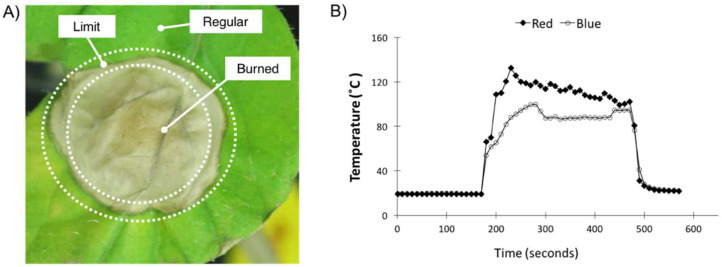
Sampling description and leaf temperature measurements. (**A**) Tomato leaf 10 days after treatment with blue (470 nm) LED light (~5000 W m^−2^, approximately 21,000 µmol m^−2^ s^−1^); the picture shows the sampled leaf areas: Burned, Limit, and Regular. (**B**) Leaf temperature measured by a thermocouple during the highlight treatment using blue and red. The light treatment started at 170 s and ended at 470 s (~5 min duration). The red LED temperature data used for comparison was published by Parrine et al. [3].

**Figure 2 life-11-00812-f002:**
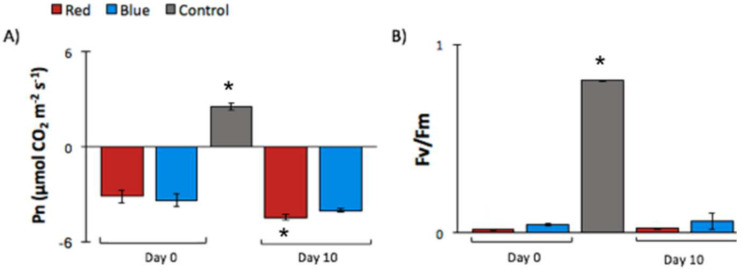
The photosynthetic parameter measurements obtained from the Burned zone of a tomato leaf after a red- or blue-light treatment (RLT and BLT, respectively). Changes in parameters from tomato plants (*Solanum lycopersicum*) stressed with deep-red (655 nm) or blue (470 nm) LEDs at ~5000 W m^−2^ intensity and control (no high-light treatment). Data points from “Day 0” were obtained immediately after the high-light treatment, and “Day 10” data points were collected 10 days after the treatment. (**A**) Net photosynthesis rate (µmol CO_2_ m^−2^ s^−1^) measured at 100 µmol m^−2^ s^−1^. (**B**) The maximum quantum efficiency of PSII photochemistry (Fv/Fm). Vertical bars indicate the ± standard error (SE) of the means (n = 3). Means are statistically significantly at *p* < 0.05 according to the Tukey’s multiple comparison tests and are indicated by *. The Pn measurement indicates the quantity of CO_2_ assimilated by the plant. The control Pn value (~2.58 µmol CO_2_ m^−2^ s^−1^) agreed with values reported in the literature at 100 µmol m^−2^ s^−1^ of light [39]. The RLT and the BLT presented negative values of Pn. The RLT had a slightly higher value for Day 0 compared to Day 10, whereas the blue sample had similar values in both data points. Although RLT had a small decrease in Pn from Day 0 to Day 10, both in BLT and RLT the Pn values indicated the driving of energy to respiration, rather than photosynthesis.

**Figure 3 life-11-00812-f003:**
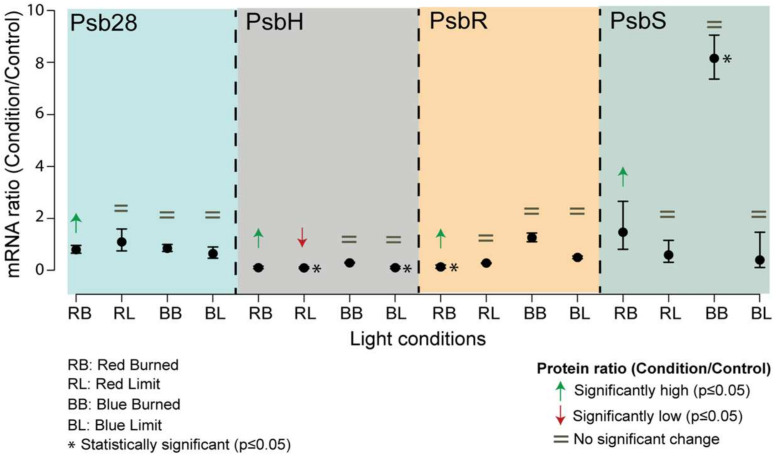
Comparison of mRNA and protein levels of different genes of interest. mRNA fold changes of genes of interest (PsbH, PsbR, Psb28, and PsbS) compared to their protein abundances. The mRNA levels were measured by qPCR. The data were analyzed by ANOVA and a Tukey adjustment of the means was performed with a *p* = 0.05 significance level. Error bars show standard deviations with n = 3 biological replicates. Arrows indicate the direction of the significant protein fold (green, high abundance; red, low abundance), the equal sign (=) indicates no change in the protein abundance between sample and control.

**Figure 4 life-11-00812-f004:**
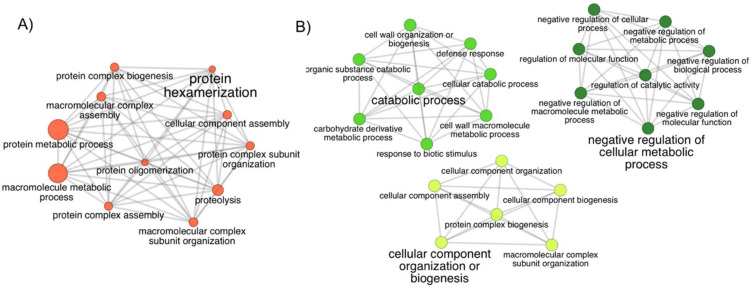
The network of gene ontology (GO) terms assigned to the proteins in high abundance in the Limit and Regular samples. Biological process GO terms assigned to proteins from the (**A**) Regular and (**B**) Limit samples. In (**B**), each of the three clusters represents terms with a closer relationship. The leading group term (biggest font size) is the term of highest significance in the network. The relationship between the terms is shown through the similarity of their assigned proteins. The node size represents the term’s statistical significance. The edges reflect the level of connectivity amongst terms, calculated by kappa statistics.

**Figure 5 life-11-00812-f005:**
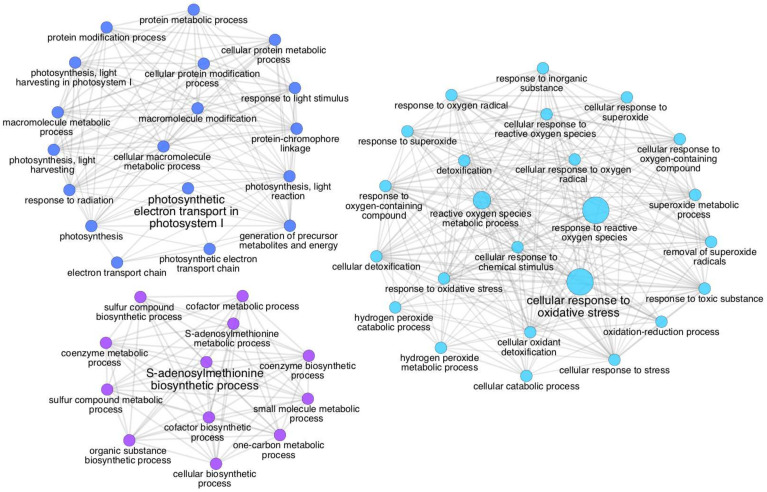
The network of gene ontology (GO) terms assigned to the proteins with low abundance in the Burned sample. The three clusters show different levels (<8) of biological process GO terms for the proteins assigned and are grouped by functional relationship. The relationship between the terms is shown through the similarity of their assigned proteins. The nodes sizes represent the term’s statistical significance. The leading group term (biggest font size) is the term of highest significance in the network. The edges reflect the level of connectivity amongst terms, calculated by kappa statistics.

**Table 1 life-11-00812-t001:** The DNA primers for RT-qPCR used in this study.

Gene	Forward Primer	Reverse Primer	PCR Product Size (bp)	Source
psb28	CCTCGCTCTCTTCTCGGAAT	GCAAAACGCGAACGGGATAG	98	This study
psbS	GGAATTGGCTTCACTAAGCA	AGTGGCTCTGCTTCATAGAT	155	This study
psbH	TCTGGTCCAAGACGAACTGC	CAAAGGGGTAGTTCCCCACC	93	This study
psbR	CAGGAAGCCCAAGGGAAAGG	GTCACCGCCCATATGGCTAA	153	This study
TPP2Acs	CGATGTGTGATCTCCTATGGTC	AAGCTGATGGGCTCTAGAAATC	149	Løvdal and Lillo [34]
Clat	ATGCAATCACACCAGCAC	ACTCAGCACAACAACAAAGG	61	Dekkers et al. [35]

**Table 2 life-11-00812-t002:** The abundance of proteins related to photosynthesis after red and blue high-light LED treatments. Values of the red and the blue light treatment are reported in log_2_-fold change (log_2_FC). Statistical tests were carried out with an FDR-controlled approach (*p* < 0.05) and the cut-off for high or low abundance was >0.7-fold change for the red light treatment dataset, and >2.5-fold change for the BLT dataset. (*) Indicates statistical significance. ^a^ Data from a previous publication [3].

	BLT (log_2_FC)			RLT ^a^ (log_2_FC)			Location	Function
Protein	Limit	Burned	Regular	Limit	Burned	Regular		
Psb28	0.02	−1.27	0.14	0.014	1.084 *	−0.044	OEC, binds to cytochrome b559	PSII assembly factor
PsbS	1.45	−0.34	0.15	0.516	1.322 *	0.322	LHCII	NPQ relaxation process
PsbH	0.73	−0.23	0.74	−0.713 *	1.646 *	0.536	PSII complex core	Electron transfer between QA and QB
PsbR	1.24	−0.12	1.52	−0.168	1.111 *	0.151	OEC, binds to PsbQ and PsbP	OEC formation

**Table 3 life-11-00812-t003:** Upregulated proteins identified in the blue light treatment dataset.

Sample Name	Protein Accession	Description	Present in BLT and RLT
Regular	Q10712	Leucine aminopeptidase 1, chloroplastic	X
	P25306	Threonine dehydratase biosynthetic, chloroplastic	X
	K4CVX0	Uncharacterized protein	X
	Q5UNS1	Arginase 2	X
	K4CVX6	Uncharacterized protein	X
Burned	K4ATA4	Uncharacterized protein	X
Limit	Q10712	Leucine aminopeptidase 1, chloroplastic	X
	K4CWC4	PR10 protein	X
	K4CVX0	Uncharacterized protein	X
	P25306	Threonine dehydratase biosynthetic, chloroplastic	X
	Q01413	Glucan endo-1,3-β-glucosidase B	X
	K4CVQ7	Uncharacterized protein	X
	Q05539	Acidic 26 kDa endochitinase	
	K4B0B4	Uncharacterized protein	
	K4C3T2	Uncharacterized protein	X
	A0RZD0	Inducible plastid-lipid associated protein	X
	Q9LEG1	Cathepsin D Inhibitor	

**Table 4 life-11-00812-t004:** Downregulated proteins identified in the blue light treatment dataset.

Sample Name	Protein Accession	Description	Present in BLT and RLT
Regular	K4CAE2	Uncharacterized protein	
	K4ASV2	ATP-dependent Clp protease proteolytic subunit	
	K4B7W7	Uncharacterized protein	X
	K4CMI6	Uncharacterized protein	X
Burned	K4CVQ7	Uncharacterized protein	X
	K4BM57	Uncharacterized protein	X
	K4BVE2	50S ribosomal protein L31	X
	P37218	Histone H1	X
	K4AYJ8	Uncharacterized protein	X
	K4B0G3	Uncharacterized protein	X
	K4AX22	Superoxide dismutase [Cu–Zn]	X
	K4C998	Uncharacterized protein	X
	P04284	Pathogenesis-related leaf protein 6	X
	E5KBY0	Snakin-2	X
	Q2MI49	Photosystem I iron-sulfur center	X
	K4C1V2	Uncharacterized protein	
	K4CX44	Uncharacterized protein	
	Q3I5C4	Cytosolic ascorbate peroxidase 1	
	K4BJY6	Uncharacterized protein	
	P43282	S-adenosylmethionine synthase 3	
	C0KKU8	Lipoxygenase	
	P10708	Chlorophyll a-b binding protein 7, chloroplastic	
Limit	K4BVE2	50S ribosomal protein L31	
	K4BX19	Uncharacterized protein	X
	K4C1V2	Uncharacterized protein	X
	K4BLU6	Uncharacterized protein	
	K4D2D7	Uncharacterized protein	X

## Data Availability

The data that support the findings of this study are available upon request from the corresponding author (M.L.).

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
