# Peer review of "Color-Specific Recovery to Extreme High-Light Stress in Plants"

_life, 2021, doi:10.3390/life11080812_

Round 1
Reviewer 1 Report
Dear Authors,
thank you for your interesting manuscript.
I think with that some parts of the manuscript should be more targeted and free general information. At times, the manuscript resembles a textbook (e.g. l.72 - l. 87). Sometimes whole page from one reference (whole caption 4.2), discussion is very poor or missing. Generally, the references needs to be up to date - zero from 2019, 2020, 2021.
Why you use photoinhibiton in keywords? No relevant issue in manuscript...
The manuscript is not balanced, resp. does not match title. It focuses more on gene, protines than on light stress in plants.
Add more information about the pathways (vacuolar) of plants under stress (light, heat etc.) - I recommended add: 10.1111/tpj.15253; 10.1007/s11738-016-2113-y; 10.1007/s11120-020-00810-2; 10.1016/j.jhazmat.2020.123256; 10.1038/s41598-021-85649-w; 10.32615/ps.2019.179
The discussion section needs to be revised. Arguments require a clearer and more accurate presentation. The understanding of plant stress response mechanisms is limited because it is limited to works that have a specific view and deliberately ignore alternatives, and do not represent a balanced view of the evidence.
References are not by style of journal.
Fig. 2 - some units missing
Fig. 4-5 - better presentation, too small - not readable - what is their meaning?
Author Response
- At times, the manuscript resembles a textbook (e.g. l.72 - l. 87).
Changes were made to the text (see manuscript, l.72 – 75).
- Sometimes whole page from one reference (whole caption 4.2),
The discussion (including section 4.2) was re-written, and in the current version, new references were added to section 4.2.
- Generally, the references needs to be up to date - zero from 2019, 2020, 2021.
We added new references from >2020:
- Espinoza-Corral R, Schwenkert S, Lundquist PK. Molecular changes of Arabidopsis thaliana plastoglobules facilitate thylakoid membrane remodeling under high light stress. Plant J. 2021 Mar 30. doi: 10.1111/tpj.15253. Epub ahead of print. PMID: 33783866.
- Li, D., Wang, M., Zhang, T. et al.Glycinebetaine mitigated the photoinhibition of photosystem II at high temperature in transgenic tomato plants. Photosynth Res 147, 301–315 (2021). https://doi.org/10.1007/s11120-020-00810-2
- Hajihashemi, S., Brestic, M., Kalaji, H.M., Salicky, M., Noedoost, F., 2020. Environmental pollution is reflected in the activity of the photosynthetic apparatus. Photosynthetica 58, 529-539. DOI: 10.32615/ps.2019.179
- Bednarczyk, D., Aviv-Sharon, E., Savidor, A., Levin, Y., & Charuvi, D. (2020). Influence of short-term exposure to high light on photosynthesis and proteins involved in photo-protective processes in tomato leaves. Environmental and Experimental Botany, 179, 104198. https://doi.org/https://doi.org/10.1016/j.envexpbot.2020.104198
- Hao, Y.F., Feng, Y.Y., Cai, L.J. et al.Effect of ABA on Photosynthesis and Chlorophyll Fluorescence in Emmenopterys henri under High Light. Russ J Plant Physiol 68, 510–518 (2021). https://doi.org/10.1134/S1021443721030067
- Shim, J.S., Park, SH., Lee, DK. et al.The Rice GLYCINE-RICH PROTEIN 3 Confers Drought Tolerance by Regulating mRNA Stability of ROS Scavenging-Related Genes. Rice 14, 31 (2021). https://doi.org/10.1186/s12284-021-00473-0
- Buccitelli, C., Selbach, M. mRNAs, proteins and the emerging principles of gene expression control. Nat Rev Genet21, 630–644 (2020). https://doi.org/10.1038/s41576-020-0258-4
- Górecka, M, Lewandowska, M, Dąbrowska-Bronk, J, et al. Photosystem II 22kDa protein level - a prerequisite for excess light-inducible memory, cross-tolerance to UV-C and regulation of electrical signalling. Plant Cell Environ. 2020; 43: 649– 661. https://doi.org/10.1002/pce.13686
- Why you use photoinhibiton in keywords? No relevant issue in manuscript
Keyword was removed.
- The manuscript is not balanced, resp. does not match title. It focuses more on gene, protines than on light stress in plants. Add more information about the pathways (vacuolar) of plants under stress (light, heat etc.) - I recommended add: 10.1111/tpj.15253; 10.1007/s11738-016-2113-y; 10.1007/s11120-020-00810-2; 10.1016/j.jhazmat.2020.123256; 10.1038/s41598-021-85649-w; 10.32615/ps.2019.179
References: 10.1111/tpj.15253; 10.1007/s11738-016-2113-y; 10.1007/s11120-020-00810-2 and 10.32615/ps.2019.179 were added.
- The discussion section needs to be revised. Arguments require a clearer and more accurate presentation. The understanding of plant stress response mechanisms is limited because it is limited to works that have a specific view and deliberately ignore alternatives, and do not represent a balanced view of the evidence.
The discussion was re-written, following the suggestions of the reviewer.
- References are not by style of journal.
The references were modified to fit the journal's style.
- 2 - some units missing
Fv/Fm and NPQ are unitless parameters (please refer to LICOR’s manual: https://www.licor.com/documents/ajncmgt9xtonajwvs3n6hxxw5u9dlfai). The information on how NPQ is calculated was added to the figure’s subtitle. Furthermore, the meaning of these parameters are explained in lines 324-346 of the manuscript.
- 4-5 - better presentation, too small - not readable - what is their meaning?
We increased the size of the figure. The figure shows the main functions of the more and less abundant proteins from the Burned sample and in high abundance in the Limit and Regular samples from the blue light experiment. The relationship between the terms is shown through the similarity of their assigned proteins. The nodes sizes represent the term's statistical significance. The leading group term (in bigger font size) is the term of the higher significance of the network. The edges reflect the level of connectivity amongst terms, calculated by kappa statistics.

Reviewer 2 Report
The manuscript of Parrine et al. deals with the consequences of extreme high light exposure on the photosynthetic activity and proteomics of tomato leaves. They conclude that high-light stress (photoinhibition) is color specific. The lacks physiological significance, and obviously lacks basic knowledge of the relevant literature regarding photoinhibition and non-photochemical quenching. Therefore the manuscript is absolutely unsuitable for publication.
Problems:
1, Lack of physiological relevance.
High-light stress when induced under natural conditions means exposure of plants up to 2000-2500 umole photons/m2/s, which might result in leaf temperatures up 30-40 C. In contrast, the applied illumination was 10 times higher of the maximal natural levels (2000-2500 umole photons/m2/s), which induced 100-120 C leaf temperatures. The illuminated leaf area was completely burned out lacking practically any photosynthetic activity. This is no wonder since the water-oxidizing complex, which is the source of electrons for photosynthetic electron transport in PSII is inactivated at around 40 C, while Rubisco, the key enzyme of CO2 fixation is inactivated at also at 40-50 C.
2, Problems with determination of NPQ.
The manuscript wants to conclude about NPQ (differential effect caused by killing the leaf with blue and red light). However, there is something basically wrong with the NPQ data. From Fig. 2C the NPQ in the control leaves is ca. 0.04. This is orders of magnitude lower than that reported in the literature: 2-4 in WT Arabidopsis (Ruban, Biochimica et Biophysica Acta 1817 (2012) 977–982); ca. 2 in tomato (Moon RY et al. Radiat. Res., 52, 238–248 (2011); ca. 1 in tomato grown under LED light (Olvera-Gonza ́lez et al. Plant Growth Regul (2013) 69:117–123) just to name a few. Therefore, the whole argumentation about NPQ values is meaningless. NPQ is calculated from variable Chl traces, so it is no wonder that from dead leaf wrong values are obtained. However, this does not explain the very small control value, so there should be some basic problem with measurement.
3, Lack of knowledge of relevant literature. One of the „discoveries” of the work is that the mechanism of photoinhibition is color specific. This phenomenon has been discovered in the mid 1970-ies and has been studied extensively by determining the action spectra of photoinhibition. The manuscript does not even mention these studies (a good summary of this topic is by Zavafer et al. (J. Photochem. Photobiol. B: 152, 2015, 247-260).
4, Proteomics. The proteomic part attempts to provide information on the recovery process of PSII after the very harsh light and heat treatment. Again, this part lacks significant scientific value since the PSII repair process is very well studied, following much milder initial damage. It does not really make much sense to look at the biogenesis of the photosynthetic apparatus (and of the whole leaf) after being burnt by 2000-2500 umole photons/m2/s) and 100-120 C leaf temperatures.
Author Response
1) We do not intend to replicate a natural condition, we are aiming to analyze how plants are impacted by extreme conditions, recovering, for example from a complete destruction of the OEC or the PSII complex. The methodology that we used shows our intentions as we analyze not only the zone of the leaf that was completely burnt, but as also the areas surrounding that zone. This way we could see differences in the recovery of the tissue in a more special approach. There are many studies showing the behavior of plants under natural conditions, or levels of high-light stress that are naturally happening on Earth. We wanted to explore beyond these limitations, since we had the meaning to do so.
2) The publications mentioned by the reviewer show a long curve comparing the values of NPQ, since they are comparing the stabilization of NPQ through time. In our approach, we compare the same data point, meaning, the value of a specific time in the curve, for all the experiments. We are comparing the same data point accross the samples in a smaller fraction of time than hours, for each one of the samples and control.
3) The objective of this study was to contribute to knowledge on light stress by studying an “unknown condition” (extreme high-light stress). This study completes our already published manuscript, by comparing another wavelength and adding more insight to the consequences of such harsh conditions. We made changes in the manuscript, as suggested by the reviewer to make our objective and the context of our research clearer.
4) The objective of the proteomics section was to determine, in depth, all changes generated by the extreme light treatment, not only on the PSII repair, and the proteins of interest, but also at many pathways. We generated a protein quantification analysis of thousands of proteins, showing how the treatment can impact their abundance. Furthermore, we analyzed by functional enrichment, what pathways are perturbed in the different sections of the treated leaf, showing the different levels of recovery.
Reviewer 3 Report
Brief summary:
Interesting manuscript that presents changes in protein and RNAm levels in response to high doses of monochromatic, red and blue radiation, from the proteomic optics. Introduction, adequate. Material and Methods thoroughly described and allow to acceptable reproducibility. Results present information on various proteins related to the metabolic activity of photoreceptors and very interesting networks of gene ontology. Conclusions are consistent with the physiology of blue light photoreceptors and stress-induced senescence.
Broad comments
Results are quite poor in the description of Sections 3.3 and 3.4, which are the basis of the manuscript. Both sections should be thoroughly reviewed. Discussion is interspersed with the results so that the specific section is poor. These paragraphs should be reassigned to Discussion. Conclusions are consistent and consistent with the study, however, culminating in a hypothesis that should be explained at the end of introduction, together with the objectives of the work.
Specific comments
Line 355-357. Data presented on line 355 are not listed on Table 2.
Line 358. The same can be said for PsbR
Line 384. You should use the term “Regular” when referring to samples, so as not to mislead with “Control” treatment
Lines 400-402. Paragraph not understandable, clarify the description.
Lines 406-408. The description of the behaviour does not match the data in Table 2 or Fig. 3.

Author Response
BROAD COMMENTS
To aid the visualization of the changes in the text, all changes were marked on the manuscript by using the tool "Track Changes".
- Results are quite poor in the description of Sections 3.3 and 3.4. Both sections should be thoroughly reviewed (...).
Sections 3.3 and 3.4 were modified, following the suggestions of the reviewer.
- Discussion is interspersed with the results so that the specific section is poor. These paragraphs should be reassigned to Discussion.
We modified the Discussion and the Results sections as suggested by the reviewer.
- Conclusions are consistent and consistent with the study, however, culminating in a hypothesis that should be explained at the end of introduction, together with the objectives of the work.
We modified the introduction to account for the suggestions of the reviewer.
SPECIFIC COMMENTS
- Line 355-357. Data presented on line 355 are not listed on Table 2.
Data is now presented in log2 fold change (as it is shown in Table 2).
- Line 358. The same can be said for PsbR.
Modified in the text.
- Line 384. You should use the term “Regular” when referring to samples, so as not to mislead with “Control” treatment
In this case, it is the control sample, and not the ‘Regular’ sample that was analyzed with qPCR, so we could generate a relative quantitative analysis.
- Lines 400-402. Paragraph not understandable, clarify the description.
We revised the paragraph and changes are showed in the text.
- Lines 406-408. The description of the behaviour does not match the data in Table 2 or Fig. 3.
We revised the text and changes are showed in the text.
Round 2
Reviewer 1 Report
Dear authors, thank you for your comments - most of them were reflected in the manuscript and contributed significantly to its improvement. However, there is still an opportunity to add relevant references on the topic - see previous revisions.
Author Response
Dear reviewer,
Thank you for your valid comments. We did not cite the remaining publications from your last review since they were diverging from the main topic of this paper. However, we would be glad to include citations of other relevant publications that you might recommend.
Reviewer 2 Report
The revision of the manuscript of Parrine et al. resulted practically no improvement. The work lacks physiological significance, and obviously lacks basic knowledge and understanding photoinhibition and the phenomenon of non-photochemical quenching. Therefore, the manuscript is unsuitale unsuitable for publication.
Problems:
1, Physiological relevance.
The treatment which was applied to the plants in not photoinhibition, but burning the plant tissue with 21 000 umole photons/m2/s light intensity (ca. 10 times higher of the maximal sunlight exposure at the surface of the Earth). This light exposure induces 100-120 C leaf temperatures, which is again well above the physiologically tolerable temperature range.
Therefore, the study has nothing to do with photoinhibition, and it is highly misleading to present the data as a result of a photoinhibition study.
2, As I pointed out in my original review the NPQ values in control leaves should be in the 2-4 range. In contrast the data presented in this study indicate a ca. 0.04 value for NPQ (Fig. 2C).
These data are meaningless.
Unfortunately, the answer of the authors for this question is also meaningless. They do not seem to understand the problem, that the values they are reporting for the control (and the treated) leaves are 50-100-fold smaller than they should be on the basis of all available literature. This shows a basic problem with the measurements, and also the lack of knowledge about the phenomenon they want to study.
3, In summary, this work:
i, lacks any physiological relevance
ii, has nothing to do with photoinhibition
iii, the presented NPQ data are technically wrong
Therefore, the work is unsuitable for publication in the present form (referring to photoinhibition and NPQ).
The only solution could be if the authors leave completely out the NPQ data and do not mention the word “photoinhibition” at all, and try make out something on the basis of their proteomic data for the process of recovery of leaves after the sh*t was burned out from them by extreme light intensity and over boiling temperatures (which looks like a good topic for an Ig Nobel contest).
Author Response
Dear reviewer,
As per your suggestions, we shifted the focus of the manuscript from photoinhibition (and NPQ disparities) to the results from the proteomics data. We removed discussions around photoinhibition and NPQ, only mentioning the concept when talking about the function of the key proteins and the temperature measurement, which was done during the light treatment, and therefore when the plant was prompt to photoinhibition.
Reviewer 3 Report
Changes proposed in rev 1 have been adequately considered and corrected. The authors have considerably improved the methodological description and discussion of the results.
Author Response
Dear reviewer,
We appreciate your insight to our manuscript and we thank you for your suggestions.